# Crimean-Congo hemorrhagic fever virus antibody prevalence in Mauritanian livestock (cattle, goats, sheep and camels) is stratified by the animal's age

Ansgar Schulz[1], Yahya Barry[2], Franziska Stoek[1], Aliou Ba[2], Jana Schulz[1], Mohamed L. Haki[2], Miriam A. Sas[1], Baba A. Doumbia[3], Peter Kirkland[4], Mohamed Y. Bah[3], Martin Eiden[1], Martin H. Groschup[1] *

1 Friedrich-Loeffler-Institut, Institute of Novel and Emerging Infectious Diseases, Greifswald-Insel Riems, Germany, 2 L'Office National de Recherche et de Développement de l'Elevage (ONARDEL), Nouakchott, Mauritania, 3 Ministère du Développement Rural, Nouakchott, Mauritania, 4 Elizabeth Macarthur Agriculture Institute, Menangle, Australia

* martin.groschup@fli.de

## Abstract

Crimean-Congo hemorrhagic fever virus (CCHFV) is one of the most widespread zoonotic arthropod-borne viruses in many parts of Africa, Europe and Asia. It belongs to the family of *Nairoviridae* in the genus of *Orthonairovirus*. The main reservoir and vector are ticks of the genus *Hyalomma*. Livestock animals (such as cattle, small ruminants and camels) develop a viremias lasting up to two weeks with absence of clinical symptoms, followed by seroconversion. This study was carried out to assess risk factors that affect seroprevalence rates in different species. In total, 928 livestock animal samples (cattle = 201; sheep = 247; goats = 233; camels = 247) from 11 out of 13 regions in Mauritania were assayed for CCHFV-specific immunoglobulin G (IgG) antibodies using enzyme-linked immunosorbent assays (ELISA) (including a novel indirect camel-IgG-specific CCHFV ELISA). Inconclusive results were resolved by an immunofluorescence assay (IFA). A generalized linear mixed-effects model (GLMM) was used to draw conclusions about the impact of certain factors (age, species, sex and region) which might have influenced the CCHFV antibody status of surveyed animals. In goats and sheep, about 15% of the animals were seropositive, whereas in cattle (69%) and camels (81%), the prevalence rate was significantly higher. On average, cattle and camels were up to twice to four times older than small ruminants. Interestingly, the seroprevalence in all species was directly linked to the age of the animals, i.e. older animals had significantly higher seroprevalence rates than younger animals. The highest CCHFV seroprevalence in Mauritania was found in camels and cattle, followed by small ruminants. The large proportion of positive animals in cattle and camels might be explained by the high ages of the animals. Future CCHFV prevalence studies should at least consider the age of surveyed animals in order to avoid misinterpretations.

**Data Availability Statement:** All relevant data are within the manuscript and its Supporting Information files.

**Funding:** This work was funded by the German Office for Foreign Affairs (German Partnership Program for Biosecurity, OR12-370-43 BIOS Subsahara), the Deutsche Forschungs Gemeinschaft (DFG) (GR980/4-1 AOBJ 630130), as well as the European Union (LEAP-AGRILEARN O1DG 18024). All funding sources were raised by M. H. G. The funders had no role in study design, data collection and analysis, decision to publish, or preparation of the manuscript.

**Competing interests:** The authors have declared that no competing interests exist.

## Author summary

Crimean-Congo hemorrhagic fever virus (CCHFV) is a silent threat that repeatedly causes severe hemorrhagic disease in humans who have been in close contact with livestock of endemic countries. The detection of CCHFV IgG antibodies in livestock can be a first indication whether the virus circulates in a given region and is thus a valuable diagnostic tool for determining the endemic status. Interestingly, earlier data from Mauritania showed a noticeable difference in IgG prevalence between sheep (18%) and cattle (67%). In contrast to sheep and cattle, current monitoring data on CCHFV IgG antibody presence in camels and goats in Mauritania is very limited. This study was conducted to provide a comprehensive up-to-date overview of CCHFV seroprevalences in the four most important Mauritanian livestock species (cattle, sheep, goats and camels). It attempts to highlight the role of potential risk factors responsible for deviating prevalences. In addition, we developed a camel-specific IgG ELISA, which can be used in future CCHFV seroprevalence studies. Furthermore, findings of this study contribute to a better understanding of the current epidemiological CCHFV situation in sub-Saharan Africa and which role different livestock species play regarding the viral circulation in endemic regions.

## Introduction

Crimean-Congo hemorrhagic fever virus (CCHFV) is one of the most widespread zoonotic arthropod-borne viruses distributed in many parts of Africa, Europe and Asia [1,2]. It belongs to the family of *Nairoviridae* in the genus of *Orthonairovirus*. Many livestock species like cattle, goats, sheep or camels can become infected with this virus and even develop viremia, but still do not showing clinical symptoms [3]. Humans can be infected by contact to infectious blood, tissue or other body fluids from viremic animals or patients. Nevertheless, most virus infections are caused by bites of infected *Hyalomma* ticks, which are the main reservoir and transmission vector of CCHFV [4]. In contrast to animals, infected people can suffer from severe symptoms, including hemorrhagic fever with case fatality rates ranging from 5% in Turkey [5] and up to 80% in China [6]. Livestock farming plays an important role for the income of the local population in Mauritania and represents an integral part of the Mauritanian economy [7]. The close contact between farmers and their animals, as well as insufficient medical or veterinary care in rural areas, bears a serious health risk for humans and animals. The first human case of CCHF in Mauritania was described in 1983 [8], triggering a first serological study for CCHFV antibodies in humans, cattle and rodents [9]. The first larger serological study was conducted by Gonzalez, LeGuenno [10] in sheep a prevalence of 18% was obtained. Another albeit much smaller study on sheep and goats in 2003 revealed a similar prevalence [11]. In 2013, in a first comprehensive study, cattle were tested solely for CCHFV IgG antibodies and a surprisingly high seropravalence of 67% was observed [12]. Investigations of cattle in Sudan [13] showed that collection site, age, husbandry system and tick infestation are the biggest risk factors for CCHFV seroprevalence. In addition, an age-related increase in CCHFV IgG antibody prevalence was already also observed in sheep [14] and cattle [13,15,16]. Therefore, this current study in Mauritania focused on potential risk factors (like age, species, sex and region) to reveal their impact on CCHFV seroprevalence rates in different livestock species. It provides a comprehensive overview of the current CCHFV IgG antibody circulation in Mauritania. We used already established tests for cattle, sheep and goats (in-house/adapted commercial assays) as well as a newly developed camel-IgG specific ELISA. Emphasis was

particularly laid on previously understudied species in Mauritania (CCHFV infections in camels and goats).

## Material and methods

### Ethics statement

The samples were taken by ONARDEL (Office National de Recherche et de Développement de l'Elevage) in order to fulfill its governmental mandate to conduct livestock animal monitoring and surveillance programs for veterinary and zoonotic pathogens following all relevant national as well as international regulations and according to fundamental ethical principles.

### Sampling sites and serum sample collection

Mauritania, located in West Africa south of the Western Sahara, has a size of 1,030,000 km$^2$ and is one of the most sparsely populated countries in Africa due to the prevalent Saharan landscape. It is dominated by a very dry, hot and windy climate. Notable amounts of rainfall in terms of a rainy season occur only in the most southern border regions of the country, which extend to the Sahel. Livestock farming is mainly practiced in the form of nomadic pastoralism and semi-extensive husbandry. Solely stationary, irrigated agricultural practices/stable housing systems do not play a major role and are only rarely practiced. For this reason, breeding of cattle less adapted to drought and hot climate is mainly found in the south of the country, while small ruminants and camels are also kept in the northern regions [17,18]. The country itself is divided into 13 different regions, which are subdivided in 44 departments. In 2015, serum samples were taken from cattle (n = 201), sheep (n = 247), goats (n = 233) and camels (n = 247) from 11 out of 13 regions (Fig 1 and Table 1). Samples from the capital region Nouakchott originated from a large abattoir that is connected with one of the most important livestock markets in this region. Within the regions, one to a maximum of three different flocks from local farmers were sampled. Considering the relatively small size of the sampled flocks, we decided to consolidate the animals region-wise. The sampling region "Nouakchott" constitutes an exception, since the samples were taken in the most important slaughterhouse/livestock market. Livestock from all over the country were driven up to this market/slaughter house. Given the large size of the country and the lack of infrastructure especially in rural the areas, it was not feasible to take samples from all species in each region. Besides, political conflicts in the border regions and the partly limited compliance of livestock owners also impaired the sampling framework. Therefore, a "convenience sampling" was performed. For 873 of 928 samples, the age and sex of the sampled animals were available.

### Establishment of a camel-specific in-house IgG CCHFV ELISA

A camel-specific CCHFV IgG ELISA was developed using His-tagged recombinant N-Protein of CCHFV-strain Kosovo Hoti (Accession no. DQ133507) as antigen. The same protein was used as the antigenic component in the other in-house assays for cattle, sheep and goats described in the section below. Half of the 96-Well F-Bottom microplates (Greiner Bio-One, Kremsmünster, Austria) were coated with 100 µl coating buffer (1x PBS; BSA 1%; pH 9) containing 0.2 µg of the antigen, whereas for the second half the antigen was omitted. The plates were incubated over night at 4˚C and afterwards blocked for 1 h at 37˚C with 200 µl blocking buffer (IDvet, Grabels, France). Serum samples were diluted 1:80 in serum dilution buffer (IDvet DB no. 11) and 50 µl of the dilution was applied twice each to the wells with antigen and the wells without antigen. The plates were then incubated for 1 h at 37˚C and washed three times with 250 µl/well washing buffer (PBS-Tween 0.1%). The unlabeled goat- anti-lama

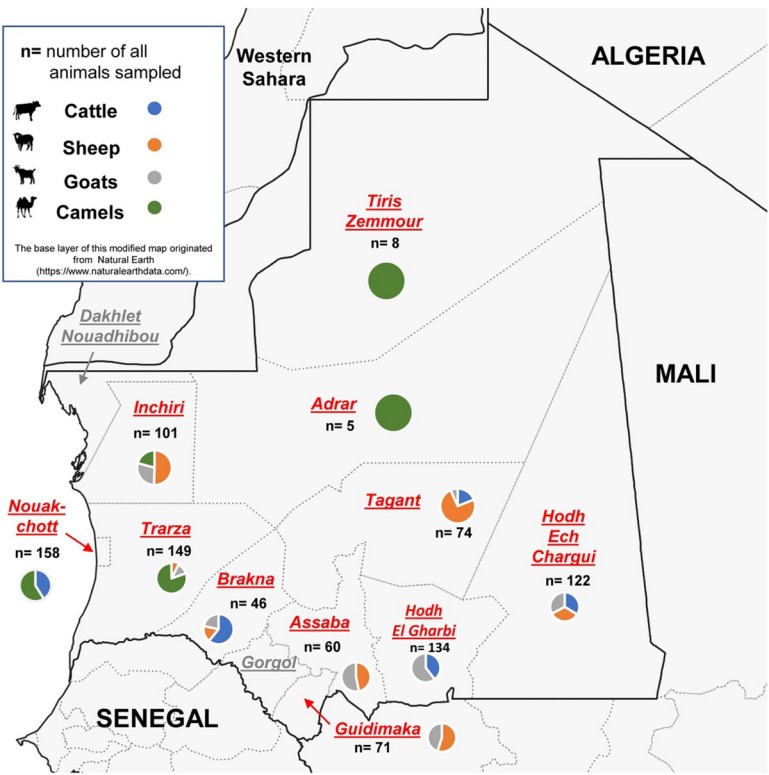

**Fig 1. Map of Mauritania which shows the proportion and total number of each different animal species sampled in each region.** All sampled regions are highlighted in red. For more detailed information on prevalence and numbers, please refer to Table 1. The base layer of this modified map originated from Natural Earth (https://www.naturalearthdata.com/).

IgG conjugate (Southern Biotech, Birmingham, USA) was diluted 1:1,500 in dilution buffer (IDvet DB no. 3), added (100 µl/well) and incubated for 1 h at 37˚C. Afterwards plates were washed three times with 250 µl/well washing buffer (PBS-Tween 0.1%). To obtain a detectable signal, a rabbit anti-goat-HRP (Southern Biotech, Birmingham, USA) conjugate in ratio 1:3,000 (100 µl/well) was used and incubated for 1h at 37˚C. Finally, the plate was washed again and 100 µl/well TMB solution (Bio-Rad, Hercules, USA) was added. After 10 min of incubation time the reaction was stopped with 100 µl 1M $H_2SO_4$ and measured with a Tecan plate reader Infinite 200 PRO (Tecan, Männedorf, Switzerland) at 450 nm against the reference wavelength of 620 nm. In order to exclude possible strong background reactions of the sera, which could falsify the result, a corrected OD (optical density) value was used (OD *(av.)* antigen−OD *(av.)* no antigen). The corrected OD value of the sample (R-sample) divided by the corrected OD value of the positive control (R-positive) gave the final result (fR) and was expressed as a percentage (fR = [R-sample/R-positive] * 100). The camel-specific in-house CCHFV IgG ELISA was validated using 42 camel sera from Australia as negative controls. According to the WHO (27 October 2019), Australia is officially free of CCHFV (https://www.who.int/emergencies/diseases/crimean-congo-haemorrhagicfever/Global_CCHFRisk_2017.jpg?ua=1). There were no field samples from Mauritanian camels available that could be considered as confirmed positive. Thus, to calculate a cut-off for the new camel-specific in-house IgG ELISA, the mean *X* and standard deviation *SD* of the corrected OD values of the negative controls were calculated and cut-off values were determined using the formula *CutOff = a·X +f·SD*, with *a* and *f* arbitrarily defined multipliers [19]. The repeatability of the test was

**Table 1. Results of seroepidemiogical studies in cattle, goats, sheep and camels in Mauritania.**

| Region | Cattle | | Goats | | Sheep | | Camels | | |
|---|---|---|---|---|---|---|---|---|---|
| | p/n | prev. (%) | p/n | prev. (%) | p/n | prev. (%) | p/i/n | prev. (%) | inc. (%) |
| **Adrar** | - | - | - | - | - | - | 5/0/5 | 100 | 0 |
| | | | | | | | | (49–100) | (0–52) |
| **Assaba** | - | - | 8/28 | 29 | 7/32 | 21 | - | - | - |
| | | | | (13–49) | | (9–40) | | | |
| **Brakna** | 14/28 | 50 | 2/8 | 25 | 5/10 | 50 | - | - | - |
| | | (31–69) | | (3–65) | | (19–81) | | | |
| **Guidimaka** | - | - | 2/39 | 5 | 2/32 | 6 | - | - | - |
| | | | | (1–17) | | (1–21) | | | |
| **Hodh Ech Chargui** | 39/41 | 95 | 15/41 | 37 | 6/40 | 15 | - | - | - |
| | | (83–99) | | (22–53) | | (6–30) | | | |
| **Hodh El Gharbi** | 35/53 | 66 | - | - | 17/81 | 21 | - | - | - |
| | | (52–78) | | | | (13–31) | | | |
| **Nouakchott\*** | 43/65 | 66 | - | - | - | - | 83/3/93 | 89 | 3 |
| | | (53–77) | | | | | | (81–91) | (1–9) |
| **Inchiri** | - | - | 8/51 | 16 | 2/29 | 7 | 6/5/21 | 29 | 24 |
| | | | | (7–29) | | (1–23) | | (11–52) | (8–47) |
| **Tagant** | 7/14 | 50 | 0/55 | 0 | 0/5 | 0 | - | - | - |
| | | (23–77) | | (0–6) | | (0–52) | | | |
| **Tiris Zemmour** | - | - | - | - | - | - | 7/1/8 | 88 | 13 |
| | | | | | | | | (47–100) | (0–53) |
| **Trarza** | - | - | 0/11 | 0 | 0/18 | 0 | 98/13/120 | 82 | 11 |
| | | | | (0–28) | | (0–19) | | (74–88) | (6–18) |
| **Total** | **138/201** | **69** (62–75) | **35/233** | **15** (11–20) | **39/247** | **16** (11–21) | **199/22/247** | **81** (75–85) | **9** (9–13) |

p = positive    n = number of tested individuals    i = inconclusive result (ELISA)

\* = Nouakchott, officially subdivided into three separated regions, has been treated as one region

95% confidence interval (CI %) in brackets

evaluated using the Bland-Altman statistics [20,21]. To verify whether the ELISA is able to detect specific reactions in camelid sera against CCHFV IgG antibodies, an alpaca was immunized under laboratory conditions with the same recombinant N-Protein of CCHFV-strain Kosovo Hoti used for ELISA. Blood samples, including day 0, were collected after immunization and tested in the ELISA. The Kosovo-Hoti N-protein was expressed in an *E. coli* vector. In order to exclude an unspecific immune reaction of the alpaca against possible *E. coli* contamination, the samples were tested twice. In the first approach, the N-protein was coated as described before, in the second approach the plates were covered with an *E. coli* lysate (Table 2). Additionally, 12 serum samples from German zoo animals (Bactrian camel and dromedary) were used as negative controls to validate the newly developed ELISA.

## Serological investigation of cattle, sheep, goats and camels by in-house CCHFV IgG ELISA

All samples were tested according to a flow chart [22] combining different ELISA and IFA test systems. The sera from cattle, sheep and goats were assayed (twice in case of a positive result) with the respective species-specific in-house CCHFV IgG ELISA system [23,24]. Positive findings were tested during a second test run with adapted commercial ELISAs (Vector Best,

**Table 2. Immunization results (OD value and final result in % (fR = [R-sample/R-positive] * 100)) of the alpaca in the novel camel-specific IgG ELISA using two different coating proteins (N-Protein of Kosovi-Hoti and *E. coli* lysate).**

| Protein coated | PC | NC | Alpaca | | | | | |
|---|---|---|---|---|---|---|---|---|
| | | | 0 dpi | 4 dpi | 7 dpi | 14 dpi | 21 dpi | 28 dpi |
| **N-protein** | 1,568 | -0,009 | -0,006 | 0,044 | -0,025 | 2,647 | 3,038 | 2,936 |
| | - | -1% | 0% | 3% | -2% | 169% | 194% | 187% |
| *E.coli* lysate | 0,047 | 0,101 | 0,106 | 0,136 | 0,082 | 0,095 | 0,090 | 0,033 |
| | - | 6% | 7% | 9% | 5% | 6% | 6% | 2% |

PC = positive control (camel field sample from Mauritania) dpi = days post immunization

NC = negative control (German zoo camel)

Novosibirsk, Russia) to confirm the results. In case of divergent results, an adapted commercial IFA (Euroimmun, Lübeck, Germany) was used to obtain a final result. The diagnostic approach for the camel sera is described above.

## Statistical analysis

The effects of age, species and sex on the CCHFV status were estimated using a generalized linear mixed-effects model (GLMM). The variables *age, species* and *sex* were considered as fixed effects. Including *region* as random regional effect with variance $\sigma^2_{region}$ led to the mixed model:

$$log\left(\frac{\pi}{1-\pi}\right) = \beta_0 + \beta_1 \cdot age + \beta_2 \cdot species + \beta_3 \cdot sex + b \cdot region \qquad (1)$$

where $\pi$ indicated the probability of an animal to be CCHFV positive. $\beta_i$ and $b$ were the regression coefficients. Model reduction was performed as the variable *sex* did not show a significant effect. This led to the final model:

$$log\left(\frac{\pi}{1-\pi}\right) = \beta_0 + \beta_1 \cdot age + \beta_2 \cdot species + b \cdot region \qquad (2)$$

Least-squares means were used for summarizing the effects of factors *age* and *species*. All analysis were performed in R (version 3.6.0 (2019-04-26)—"Planting of a Tree" [25] using the package lsmeans [26].

## Results

### Validation of the new camel-specific in-house IgG CCHFV ELISA

Forty-two Australian negative camel sera were used for the calculation and validation of the cut-off. Two cut-off values were calculated to define a range of inconclusive results. The upper cut-off value ($a = 1, f = 2$) was set to 19.96% and samples above this value were considered to be CCHFV positive. The lower cut-off value ($a = 1, f = 1$) was 10.45% and samples below this value were considered as negative. Samples showing OD values between these two cut-off values were considered to be inconclusive. Bland-Altman analysis [20] indicated good repeatability of the assay. However, two of the Australian sera showed a clear positive reaction in the ELISA. In the second test run, both were positive again, but deviated strongly from the first run. The rest of the samples showed no significant deviation in their values upon repetition. Since no positive reference field sera existed, a serum of an immunized alpaca was used as reference. On days 0, 4 and 7 post immunization, no specific reaction was detected by the ELISA. All sera collected after the 14th day post immunization were highly positive indicating that the ELISA is able to detect a specific seroconversion in camelids (Table 2) without any or only

minimal reactivity to *E. coli* antigens itself. In addition, 12 camel sera originating from German zoos were tested negative (S1 Table).

## Serological and statistical findings in cattle, goats, and sheep and camels

This study revealed considerable prevalence differences between animal species (Table 1 and Fig 2B). Cattle samples showed a seroprevalence of 68.66% with regional differences ranging from 50 to 95%. 16% of sheep (ranging from 0 to 50%) and 15% of goats (ranging from 0 to 37%) were positive. The prevalence of camels (81%), however, was even higher than in cattle. In total, thirty-two inconclusive sera from cattle (8/13), goats (1/13), and sheep (0/6) were additionally tested in the IFA, of which nine were finally considered positive.

Moreover, we observed that older animals were more likely to be positive than younger animals (Table 3 and Fig 2A). A more detailed statistical analysis revealed a significant effect of age and species on the CCHFV status of the investigated animals, whereas sex had no influence. The region was another reason of random variability. Table 4 shows the comparison of the different age and species categories regarding their influence on the CCHFV antibody status. All other age categories were clearly different from group A (0–2 years). In addition, the second youngest age group B (3–4 years) differed almost from all others with one exception. There was no difference between B and the three next older cohorts C (5–6 years), D (7–10 years) and E (>10 years). When considering the species, there were no significant seroprevalence differences between goats and sheep nor between cattle and camels.

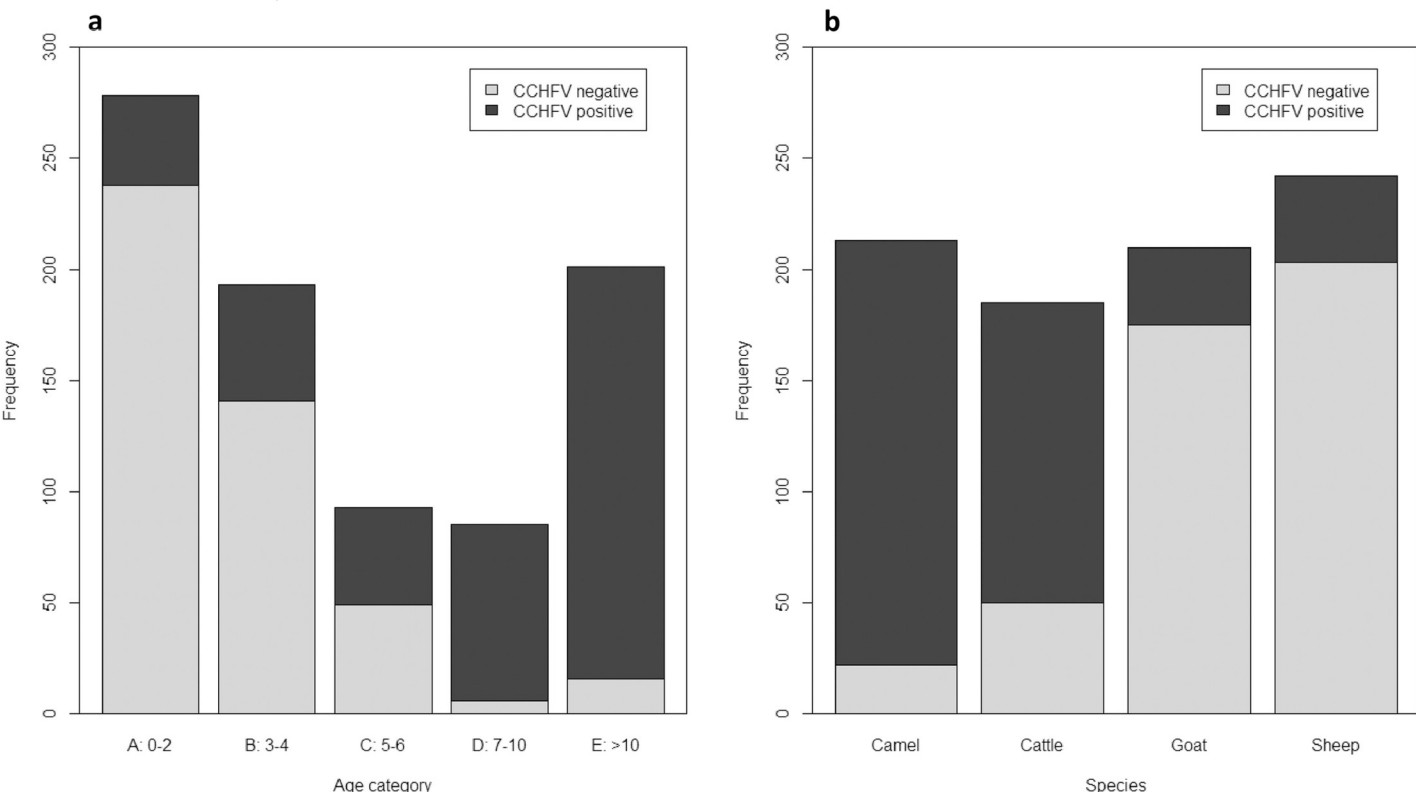

**Fig 2.** Effects of age (a) and species (b) regarding CCHFV IgG antibody status in different livestock species in Mauritania. Overall, for 850 of 928 samples, age, species, region and an unambiguous CCHFV-antibody status were available. All inconclusive results were excluded. Fig 2B summarizes the age dependency for all four species.

**Table 3. Age-related prevalence of IgG-specific CCHFV antibodies in cattle, goats, sheep and camels.**

| Age group (years) | Cattle (Ø_age: 5.84) | | Goats (Ø_age: 2.88) | | Sheep (Ø_age: 2.91) | | Camels (Ø_age: 11.55) | |
|---|---|---|---|---|---|---|---|---|
| | prev. (%) | p/n | prev. (%) | p/n | prev. (%) | p/n | prev. (%) | p/n |
| **A 0–2** | 37 | 22/60 | 8 | 8/95 | 8 | 9/116 | 25 | 2/8 |
| | (25–50) | | (4–16) | | (4–14) | | (3–65) | |
| **B 3–4** | 63 | 10/16 | 20 | 17/87 | 22 | 18/82 | 88 | 7/8 |
| | (35–85) | | (12–29) | | (14–32) | | (47–100) | |
| **C 5–6** | 92 | 23/25 | 20 | 4/20 | 28 | 12/43 | 100 | 5/5 |
| | (74–99) | | (6–44) | | (15–44) | | (48–100) | |
| **D 7–10** | 94 | 64/68 | 75 | 6/8 | 0(0–98) | 0/1 | 90 | 9/10 |
| | (86–98) | | (35–97) | | | | (55–100) | |
| **E >10** | 85 | 17/20 | - | - | - | - | 92 | 167/182 |
| | (62–97) | | | | | | (87–95) | |

prev. = prevalence        p = positive        n = number of tested individuals

95% confidence interval (CI %) in brackets

# Discussion

The aim of the study was to reveal factors, which have a decisive impact on the CCHFV sero-prevalences in different livestock species. For this purpose, several established diagnostic assays for serological IgG detection in cattle, sheep and goats were used. Since there was no diagnostic assay readily available for camels, a new camel-specific in-house ELISA was developed. With two exceptions, all negative reference sera were clearly negative. Two serological reactors from camels from Australia may either be non-specific immune responses or represent cross-reactions to other orthonairoviruses circulating in Australia. An importation history for these camels from endemic regions to Australia could be excluded. Furthermore, the detection of a distinct immune response of the immunized alpaca from the 14th day post immunization onwards supports the validity of the ELISA in terms of specific reactions (Table 2).

In general, serological testing for CCHFV antibodies can be considered as a challenging issue. As there are many different factors influencing the assay, findings should be interpreted carefully. Besides cross-reactivity with other Orthonairoviruses, one of the major challenges is posed by the large genetic variability linked to the geographical distribution of the virus [27].

**Table 4. Results of the generalized linear mixed-effects model (GLMM) for age (a) and species (b) after p-value adjustment using multivariate t-distribution.** The significance level was set to 0.05. Significant results were marked in bold.

| a | | b | |
|---|---|---|---|
| Differences between age groups | p-value | Differences between species | p-value |
| A: 0–2 –B: 3–4 | **< 0.0001** | Camel—Cattle | 0.9609 |
| A: 0–2 –C: 5–6 | **< 0.0001** | Camel—Goat | **< 0.0001** |
| A: 0–2 –D: 7–10 | **< 0.0001** | Camel—Sheep | **< 0.0001** |
| A: 0–2 –E: > 10 | **< 0.0001** | Cattle—Goat | **< 0.0001** |
| B: 3–4 –C: 5–6 | 0.5970 | Cattle—Sheep | **< 0.0001** |
| B: 3–4 –D: 7–10 | **0.0003** | Goat—Sheep | 0.9915 |
| B: 3–4 –E: >10 | **0.0458** | | |
| C: 5–6 –D: 7–10 | **0.0112** | | |
| C: 5–6 –E: >10 | 0.3709 | | |
| D: 7–10 –E: >10 | 0.7875 | | |

Currently, there is no serological assay available, which can cover this broad range of genetic diversity in terms of antigenic components used in one single test. Therefore, it has to be emphasized that our in-house assays utilized antigens from Eurasian CCHFV strains.

This study provides an updated overview of the CCHFV-IgG antibody circulation in the four major livestock species in Mauritania. Using the same assay as Sas, Mertens (12), almost an identical prevalence (69% compared to 67%) was found in cattle. The seroprevalence found in sheep (16%) deviated only slightly from older data (18.3%) of Gonzalez, LeGuenno (10). Previously, only 27 goat samples have been investigated in Mauritania within a human case report study of Nabeth, Cheikh (11), detecting three positive animals (11.1%). In this study, we detected a prevalence of 15% among 233 tested goats and thus could confirm these findings. To our knowledge, camels were never screened for CCHFV-specific antibodies in Mauritania before. Only a limited number of CCHFV seroprevalence studies in camels have been carried out so far [28], with recent findings indicating prevalences of 5.3% in Iran [29], 10.5–14.4% in Niger [30], 21.3% in Sudan [31] and 67% in the United Arab Emirates [32] respectively. Using the newly established camel-specific ELISA, this study revealed a surprisingly high proportion of positive animals (81%). More detailed surveillance data on camels are therefore needed which should take the most important risk factors (age, collection site, tick infestation and husbandry system) into account to clarify the role of camelids in the CCHFV transmission and maintenance cycles.

Age and species dependent CCHFV antibody prevalences were observed for cattle and camels, as well as for sheep and goats (Fig 2 and Table 3). Significantly higher CCHFV antibody prevalences were found in young age groups (0–2 years) of cattle and camels (37% /25%) compared to sheep and goats (8% /8%). Moreover, antibody prevalences rised in the older age groups (3–4 years and older) in all species. However, cattle and camels reached more than 80% CCHFV antibody prevalences while sheep and goats of the corresponding ages remained at about 20% respectively. Interestingly, the average age of tested cattle in Hodh el Chargui (7.24 years), which had a high prevalence of 95%, was also considerably higher than the overall average age of all examined bovines (5.84 years; Tables 1 and 3). Our findings correlate well with previous observations of age-depended seroprevalences in sheep [14] and cattle [13,15,16]. Furthermore, it could be shown that the age also has a significant influence on CCHFV seroprevalences in goats and camels (Table 3 and Fig 2A).

It is clear that the steady rise of antibody prevalences with increasing age in all species can be due to an additive effect. However, hardly any experimental data are available on the CCHFV antibody persistence in livestock and wildlife, determining the extent of this effect. Obviously, higher age coincides with a greater chance of exposure to CCHFV-positive ticks in endemic regions and thus becoming infected with the virus.

CCHFV infection studies in cattle, sheep and equids revealed that all examined species develop a short-term viremia [3]. Animal infection trials in sheep demonstrated IgG antibody titers that persisted up to 30 respectively 40 days post- infection [33,34]. On the other hand, antibody persistence of up to 256 days post infection (African hedgehog) and 512 days post infection (Cape ground squirrel) was observed in wild mammals [35]. In humans, IgG antibodies were detected up to 5 years after recovery from a CCHFV infection [36]. Unfortunately, there are no experimental data for camelids to date. Therefore, more long-term CCHFV infection studies of livestock animals are necessary to improve our understanding of the humoral and cellular immunological memory in host species.

Sheep and goats represent an important source of meat in Mauritania and are therefore slaughtered at an early age, while cattle and camels are often used as dual-purpose breeds. These species are both kept for dairy farming and sent to the slaughterhouse only when milk yields decrease. For meat production, primarily young animals are slaughtered. Due to the fact

that female camels produce the largest amount of milk between the 6th and 8th parity (i.e. 10–12 years of age) [37], they reach a high average age and are often slaughtered late at the age of 18–20 years when milk yields start to decrease gradually. This might explain why most of the camels in this study were older than 10 years. However, age effects alone cannot explain prevalence differences between large and small ruminants in young, as well as in old age groups. Therefore, differences in husbandry systems may also play a role in terms of CCHFV seroprevalence. It was shown for cattle [13] and small ruminants [38] that nomadic grazing will significantly increase seroprevalences due to higher tick exposition risks compared to stationary trough feeding systems. Vegetation, lack of tick treatment and the absence of poultry (which pick ticks) are also considered as potential risk factors [38]. In general, the husbandry of camels in Mauritania differs from the small ruminants. Camels are extensively kept and spend almost their entire live grazing in the bushlands. In contrast, sheep and goats are bred semi-extensively and spend less time on the pasture. They also receive more frequently veterinary treatment (antiparasitics etc.) than camels. Although there is no detailed information on the treatment of the sampled animals available, these parameters may also account for higher CCHFV prevalences in cattle and camels compared to sheep and goats of same age groups as well as for the deviating serological findings among the different regions.

## Supporting information

**S1 Table. ELISA results of the camel samples from Australia and German zoos used as negative reference samples.** The two Australian samples marked in yellow (Aus 26/31) are the two outliers showing a strongly deviating OD value.
(DOCX)

## Acknowledgments

We thank Martina Abs, René Schöttner, Bärbel Hammerschmidt and all animal caretakers for their excellent technical support. We also thank all zoological gardens that have kindly provided serum samples from camels.

## Author Contributions

**Conceptualization:** Ansgar Schulz, Yahya Barry, Franziska Stoek, Miriam A. Sas, Martin H. Groschup.

**Data curation:** Ansgar Schulz, Yahya Barry, Jana Schulz.

**Formal analysis:** Jana Schulz.

**Funding acquisition:** Martin H. Groschup.

**Investigation:** Ansgar Schulz, Yahya Barry, Franziska Stoek.

**Methodology:** Ansgar Schulz, Aliou Ba.

**Project administration:** Mohamed L. Haki, Baba A. Doumbia, Mohamed Y. Bah, Martin H. Groschup.

**Resources:** Baba A. Doumbia, Peter Kirkland, Mohamed Y. Bah.

**Software:** Jana Schulz.

**Supervision:** Mohamed L. Haki, Miriam A. Sas, Peter Kirkland, Martin Eiden, Martin H. Groschup.

**Validation:** Ansgar Schulz, Martin H. Groschup.

**Visualization:** Jana Schulz.

**Writing – original draft:** Ansgar Schulz.

**Writing – review & editing:** Ansgar Schulz, Franziska Stoek, Jana Schulz, Martin Eiden, Martin H. Groschup.

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
