## [Decision Letter · Decision Letter 0]

9 Dec 2020

Dear Dr. Groschup,

Thank you very much for submitting your manuscript "Crimean-Congo hemorrhagic fever virus antibody prevalence in Mauritanian livestock (cattle, goats, sheep and camels) is stratified by the animal’s age" for consideration at PLOS Neglected Tropical Diseases. As with all papers reviewed by the journal, your manuscript was reviewed by members of the editorial board and by several independent reviewers. The reviewers appreciated the attention to an important topic. Based on the reviews, we are likely to accept this manuscript for publication, providing that you modify the manuscript according to the review recommendations. 

Sincerely,

Andrea Marzi

Deputy Editor

Reviewer's Responses to Questions

**Key Review Criteria Required for Acceptance?**

**Methods**

-Are the objectives of the study clearly articulated with a clear testable hypothesis stated?

-Is the study design appropriate to address the stated objectives?

-Is the population clearly described and appropriate for the hypothesis being tested?

-Is the sample size sufficient to ensure adequate power to address the hypothesis being tested?

-Were correct statistical analysis used to support conclusions?

-Are there concerns about ethical or regulatory requirements being met?

Reviewer #1: Camel ELISA is well validated. Positive samples were confirmed in repeat measurements. Samples collected from across the country.

Reviewer #2: Please provide more detailed information regarding the sampling framework in your Methods section:

What considerations were made in determining the sample sizes? In each geographic region, how were individual animals chosen for sampling? Were they all sampled from a single herd/farm? Why were some animals only sampled from certain regions and not others?

Why was CCHFV strain Hoti utilized as the antigenic component of the Camel ELISA? Why not an African strain like IbAr10200 or UG3010? It would be nice to see a comment on this because some data suggest that even NP-based ELISA's for CCHF may lose sensitivity based on antigenic (geographic) variation (PMID: 23171700).

**Results**

-Does the analysis presented match the analysis plan?

-Are the results clearly and completely presented?

-Are the figures (Tables, Images) of sufficient quality for clarity?

Reviewer #1: Results are clearly presented. 

Authors could consider presenting their seroprevalence data on the map of figure 1 to give a better spatial understanding of the data.

Reviewer #2: Please provide more detailed information/legend for Figure 1 (see Methods section comments).

**Conclusions**

-Are the conclusions supported by the data presented?

-Are the limitations of analysis clearly described?

-Do the authors discuss how these data can be helpful to advance our understanding of the topic under study?

-Is public health relevance addressed?

Reviewer #1: Conclusions are well supported, authors discuss data in context of other data thoroughly and provide discussion on potential explanations for their findings.

Reviewer #2: The Discussion section references previous studies that have shown nomadic grazing can significantly impact CCHF seroprevalences (Lines 312-313). Can you comment on how this relates to your dataset (or not)? Please describe the type of livestock farming practices that are used in Mauritania (nomadic pastoralism? stationary, irrigated agricultural practices? mix of both?). Do these practices vary by the regions you sampled? 

Lines 314-316: You mention vegetation or lack of tick treatments as possible parameters that could account for higher CCHFV prevalence in cattle/camels vs sheep/goats. Do you have any additional information to help lend support to these hypotheses? Is it possible comment on differences in vegetation/agroecological zones (AEZs) between sample sites or regions? What about differences in acaricide practices?

Some of the regional differences in CCHFV seroprevalence are interesting, for example, cattle in Hodh El Chargui. It would be useful to point this out & have a short discussion about why this might be.

**Editorial and Data Presentation Modifications?**

Reviewer #2: Lines 70-71: Please expand on this & briefly describe the type of livestock farming practices that are used in Mauritania. Nomadic pastoralism? More stationary, irrigated agricultural practices? or a mix of both?

Line 74 mentions a "first" CCHFV serological study on cattle in Muaritania in 1983. Then, Lines 77-78 mention that cattle were tested for IgG antibodies for the first time in 2013. This is a bit confusing. What were they measuring in the 1983 study? Just a quick specification in the text would help.

Reviewer #3: Data from negative control sera from Australia and German are not presented. These data should be include in supplement or included in the main figures or tables. Also, how many samples were considered "inconclusive" and please include the results from IFA data as well.

**Summary and General Comments**

Reviewer #1: The manuscript by Schulz et al is well written and data is clearly presented. Author's data adds to the body of knowledge on the wide prevalence of CCHFV throughout Africa but add insight onto how animal age impacts seroprevalence. Authors provide key validation data on their in-house ELISA and their conclusions are well supported. Authors discuss their results in the context of previous findings thoroughly. 

Minor Comments:

Authors found that camels even at a young age were significantly more likely to be sero-positive than goats or sheep. Authors discuss that this could be due to husbandry practices. Is it known if the local population indeed have different husbandry practices for these livestock species? In regions with multiple species, is it typical for a farmer or family to have multiple species?

Some editing for grammar and syntax is needed. Ex. Lines 294-295, 290-292

Under what context were the samples outside of the capital region collected? Markets, farms, slaughterhouses? 

How does the climate of the country differ across the regions?

Reviewer #3: This manuscript describe the development of an ELISA capable of detecting CCHFV antibody in camel. Rigorous validation was performed to ensure the robustness of the assay. A serosurvey was performed in sheep, cattle, camels and goats in several regions of Mauritania. The data analysis revealed that CCHFV seropositivity is strongly linked to age rather than species. I only have minor edits that can be addressed in a revised manuscript.

1) Abstract :"The reservoir and vector are ticks of the genus Hyalomma. Livestock animals (such as cattle, small ruminants and camels) develop a viremias..", replace "reservoir" by "main reservoir".

2) "he upper cut-off value ( = 1, = 2) was set to 19.96 % and samples above this value were considered to be CCHFV positive. The lower cut-off value ( = 1, = 2) was 10.45 % and samples below this value were considered as negative"

There seem to be a mistake in f value here?
---

## [Editor Report · Decision Letter 1]

9 Feb 2021

Dear Dr. Groschup,

We are pleased to inform you that your manuscript 'Crimean-Congo hemorrhagic fever virus antibody prevalence in Mauritanian livestock (cattle, goats, sheep and camels) is stratified by the animal’s age' has been provisionally accepted for publication in PLOS Neglected Tropical Diseases.

Best regards,

Andrea Marzi

Deputy Editor

---

## [Editor Report · Acceptance letter]

7 Apr 2021

Dear Dr. Groschup,

We are delighted to inform you that your manuscript, "Crimean-Congo hemorrhagic fever virus antibody prevalence in Mauritanian livestock (cattle, goats, sheep and camels) is stratified by the animal’s age," has been formally accepted for publication in PLOS Neglected Tropical Diseases.

Best regards,

Shaden Kamhawi

co-Editor-in-Chief

Paul Brindley

co-Editor-in-Chief
